# Assessing Preclinical Research Models for Immunotherapy for Gynecologic Malignancies

**DOI:** 10.3390/cancers13071694

**Published:** 2021-04-02

**Authors:** Jhalak Dholakia, Carly Scalise, Rebecca C. Arend

**Affiliations:** Department of Obstetrics and Gynecology, University of Alabama at Birmingham, Birmingham, AL 35294, USA; jjdholakia@uabmc.edu (J.D.); cbscalise@uabmc.edu (C.S.)

**Keywords:** gynecologic oncology, translational research, preclinical models, immunotherapy, cancer

## Abstract

**Simple Summary:**

Novel treatments in immunotherapy for gynecologic oncology have not successfully developed from preclinical research to clinical trials. Preclinical models used to investigate immunotherapy agents are summarized in order to enhance understanding of the inherent limitations and areas of improvement necessary to optimize this research. It is necessary to develop and utilize appropriate preclinical models whose outcomes can be translated to clinical practice in order to identify novel treatments to improve outcomes in patients with gynecologic malignancies.

**Abstract:**

Gynecologic malignancies are increasing in incidence, with a plateau in clinical outcomes necessitating novel treatment options. Immunotherapy and modulation of the tumor microenvironment are rapidly developing fields of interest in gynecologic oncology translational research; examples include the PD-1 (programmed cell death 1) and CTLA-4 (cytotoxic T-lymphocyte-associated protein 4) axes and the Wnt pathway. However, clinical successes with these agents have been modest and lag behind immunotherapy successes in other malignancies. A thorough contextualization of preclinical models utilized in gynecologic oncology immunotherapy research is necessary in order to effectively and efficiently develop translational medicine. These include murine models, in vitro assays, and three-dimensional human-tissue-based systems. Here, we provide a comprehensive review of preclinical models for immunotherapy in gynecologic malignancies, including benefits and limitations of each, in order to inform study design and translational research models. Improved model design and implementation will optimize preclinical research efficiency and increase the translational value to positive findings, facilitating novel treatments that improve patient outcomes.

## 1. Introduction

An Unmet Need for Novel Treatments in Gynecologic Oncology: The Potential of Immunotherapy

The increasing incidence and plateau in outcomes for gynecologic malignancies indicate a significant need for novel treatment options. Uterine cancers are the third most common malignancy for women in the United States and are increasing in incidence and prevalence. These cancers, when recurrent or advanced-stage, exhibit limited responses to existing treatments. Patients with advanced-stage and recurrent cervical cancers have a similarly poor prognosis: survival rates are largely unchanged over the last 50 years [1]. In OC (ovarian cancer), over 70% of women present at an advanced stage, and the majority will experience recurrence and die of disease [2,3,4]. OC is the most deadly gynecologic malignancy, with 13,940 deaths estimated for 2020 and a five-year overall survival rate of approximately 39% [1]. These statistics indicate an unmet need for landscape-changing therapies for gynecologic malignancies.

The potential for immunotherapy in gynecologic oncology is based on successes in renal cell carcinoma, non-small-cell lung carcinoma, and melanoma [5]. However, the field has yet to hit any “home runs”. There is interest in harnessing the enhanced sensitization of the adaptive immune system and modification of the TME (tumor microenvironment) to promote a more robust immune response. In particular, immunotherapy holds promise in changing the landscape in poor-prognostic cancer types such as ovarian epithelial carcinoma [5,6] and advanced uterine and cervical cancers. However, gynecologic cancers are characterized by many features of a “cold” system (low tumor mutational burden, poor antigenicity, low immunoscore, and lack of T-cell killing): these features make immunotherapy more challenging [7]. Thus, there is a need for an enhanced understanding of the factors contributing to these characteristics in order to identify treatment strategies.

Preclinical models are necessary for the preliminary therapeutic development of effective agents. However, preclinical immunotherapy research in gynecologic oncology has thus far shown limited translational value to the clinical setting. This may be in part due to the inherent limitations of currently used models in accurately representing the human immune system and TME. Without a thorough contextualization of these limitations, research risks premature drug acceptance and attempts at translation to human studies. In so doing, researchers may under-recognize the complexity of immune pathways and TME, setting themselves and patients up for failure. This review will provide an overview of preclinical gynecologic cancer models for immunotherapy research in order to improve study design, data collection, and analysis. We will interpret the unique aspects of each methodology with a focus on translational applicability to the clinical setting, including limitations of each model that may preclude meaningful outcomes for immunotherapy agents in human trials.

## 2. Targeted Molecular Therapies in Gynecologic Malignancies

Targeted molecular therapies have shown promise in the treatment of gynecologic malignancies and offer valuable insight into the unexpected complexity in translation from preclinical to clinical studies. An example is bevacizumab, a human monoclonal antibody targeting VEGF (vascular endothelial growth factor), originally investigated for its angiogenic influence. However, its roles in vascular permeability and cell migration support the complexity of anti-VEGF agents’ impact on the TME [8]. When added to standard of care treatment, anti-VEGF therapy improves overall and progression-free survival for OC patients with poor prognosis [9,10,11]. In cervical cancer, the addition of bevacizumab to chemotherapy in metastatic or recurrent disease improves overall survival [12]. Anti-VEGF agents provide an example of TME-directed therapy, and studies are ongoing regarding optimal use, particularly as it relates to its potential in causing hypoxia, which could sensitize cells to ICB (immune checkpoint blockade).

Similarly, PARP (poly ADP-ribose polymerase) inhibitors were originally intended as a targeted treatment for patients with germline BRCA mutations and deficiencies in homologous recombination pathways, inducing “synthetic lethality” to tumor cells via inhibition of single-strand DNA repair [13]. Patients with BRCA mutations on maintenance PARP inhibitors after standard of care primary treatment for OC experienced a 70% lower risk of progression or death [14]; in addition, patients without these mutations showed improved progression-free survival when PARP was used as maintenance therapy after a platinum-sensitive recurrence [15]. As a result, PARP inhibitor use has expanded to consideration as part of primary treatment and/or maintenance in many, if not all, patients with advanced OC [16,17], and phase I and II clinical trials are ongoing to investigate the role of PARP inhibition in endometrial cancers [18]. Recently, PARP inhibitors have been shown to activate anti-tumor immune responses via the STING (stimulator of interferon genes) pathway, potentially sensitizing tumors to ICB and expanding their role in the treatment of gynecologic malignancies [19]. Research is also ongoing regarding chemokine CCL5, which is epigenetically silenced in OC to enable immune cloaking [20], as well as preclinical and clinical investigation of the PI3K/AKT/mTOR pathway [21]. These examples show that the development of effective novel agents for gynecologic malignancies relies on understanding cancer pathophysiology at the molecular, cellular, and TME levels, underscoring the importance of effective preclinical studies.

### 2.1. Immunotherapy in Gynecologic Malignancies

At its foundation, immunotherapy seeks to create a durable immune response that acts against cancer cells, both eliminating existing disease and preventing recurrence. Cancer evasion from the immune system occurs at multiple levels, and harnessing the cancer–immunity cycle similarly may require a multi-step approach [22]. The main components of this process in gynecologic malignancy research include improving antigen detection and activation of T-cell responses, as well as manipulating the TME to promote a robust local immune response against tumor cells [23]. For example, CAR-T (chimeric antigen receptor-modified T-cell) therapy aims to genetically construct T cells to specifically target tumor antigens [24]. Clinical trials are ongoing to assess the safety and efficacy of CAR-T; however, its use is still investigational in gynecologic malignancies, as is the use of epigenetic modulators such as HDAC (histone deacetylase) and DNA methyltransferase inhibitors. Vaccination against either specific tumor antigens or whole tumor cells has been limited by an inability to recruit sufficient T-cell responses and, despite extensive preclinical research, has not shown durable responses in clinical trials [25]. The challenges faced by these approaches—successful in other cancers—in reliably stimulating a clinical response indicate the complexity of immunotherapy in gynecologic malignancies.

Assessing the TME in gynecologic malignancies, the presence of TILs (tumor-infiltrating lymphocytes) has been associated with improved progression-free and overall survival in patients with advanced OC [26]. TILs have been associated with increased local levels of immune chemokines, implying a multifaceted, robust immune response against the tumor, and tumors with increased levels show improved responses to immunotherapy and improved survival outcomes [27]. As a result, there is significant interest in manipulating the TME for targeted immunotherapy. An example of a tumor cellular factor impacting the TME and local immune response is PD-L1 (programmed cell death ligand 1), which interacts with PD-1 (programmed cell death 1) receptors on cytotoxic T cells. PD-L1 has been suggested to suppress CD8+ T cells; overexpression on tumor cells confers a poorer prognosis for patients [28]. PD-1 interactions with target ligands PD-L1 and PD-L2 also result in net decreased proliferation and survival of T cells, decreased local cytokine release, and enhanced regulatory T-cell activity, contributing to local suppression of the immune response [29]. CTLA-4 (cytotoxic T-lymphocyte-associated protein 4) interactions represent another mechanism of immune modulation under investigation. CTLA-4 activation promotes the immunosuppression of CD8+ T cells and is influenced by other T-cell receptor pathways; antibody-mediated blockade has been shown to stimulate anti-tumor immune responses [30]. Increased CTLA-4 expression results in a net inhibitory response to the T cell itself, leading to decreased T-cell survival, decreased local cytokine release (namely IL-2 [29]), and a more “pro-tumor” microenvironment. These preclinical discoveries have had translational applicability: clinical trials in both PD-1/PD-L1 and CTLA-4 immune checkpoint inhibition have been promising [31]. Anti-PD-1 treatment has shown encouraging and durable responses, including complete responses, for recurrent vulvar, vaginal, and cervical cancers [32]. Studies assessing treatments with both agents aim to alter the cancer–immunity cycle at two separate points for a synergistic impact [22]. Clinical trials have shown positive overall response rates (ORR) and modestly improved progression-free survival in epithelial ovarian, primary peritoneal, or fallopian tube cancers with poor prognosis, as well as anti-tumor effects in recurrent endometrial cancer [33,34].

The PI3K/AKT/mTOR signaling pathway is of increasing interest in gynecologic oncology. Originally investigated for its role in cell cycle progression, it also has been implicated in angiogenesis and hypoxia tolerance, facilitating tumor growth [35]. Activating mutations in this pathway (e.g., *PIK3CA*) are common in gynecologic malignancies [36,37]. PI3K overexpression is also implicated in oncogenesis, particularly in cervical cancer, and pathway inhibition has been shown to stimulate tumor cell apoptosis [38]. The PI3K/AKT/mTOR signaling pathway is also under investigation for its influence on the TME: recent studies suggest its role in T-cell recruitment and macrophage polarization [39]. Numerous ongoing phase I/II clinical trials aim to assess the role of targeting this component, both individually and in combination with hormonal therapy [40,41]; however, due to the pathway’s inherent complexity and diverse influences, more preclinical research is necessary to enhance our mechanistic understanding and inform effective use in the clinical setting.

### 2.2. Utilization of Targeted Therapies in Priming Responses to Immunotherapy in Gynecologic Malignancies

Combinations of immune checkpoint inhibitors with PARP inhibitors, chemotherapy, and anti-angiogenic therapies are an active area of research. Anti-VEGF agents are now postulated to positively impact T-cell trafficking and PD-L1 expression, supporting a possible role in the TME. Similarly, PARP inhibitors may enhance TME interferon pathways and immune cell infiltration. For example, combination treatment with PARP inhibitor and PD-1 inhibitor indicates an ORR of 18% with a disease control rate of 65%, improved from ORR with monotherapy [42,43,44,45,46]. Similarly, PARP inhibition with an anti-PD-L1 agent showed a 68% ORR, with six complete responses [47]. Anti-VEGF and anti-PD-L1 combination therapy also improved ORR in recurrent OC compared to anti-PD-L1 alone, suggesting a possible synergy [48,49,50]. In patients with advanced endometrial cancer, anti-PD-1/anti-CTLA-4 combination therapy demonstrated a survival benefit compared to chemotherapy, contributing to FDA approval for the regimen [51]. Combination anti-CTLA-4 and PARP inhibition has also demonstrated survival benefits in preclinical models [52]. Despite these clinical trial results, no preclinical data regarding the timing of administration have been published. Preclinical research into the impact of medication timing (e.g., sequential or concurrent) could inform and enhance this observed synergy. In addition to these efforts, there is active research using other agents, such as TGFβ, epigenetic therapy, and the Wnt signaling pathway [53].

Preclinical research has the opportunity to identify these interactions, enabling molecular targeting and synergistic treatment, but it requires accurate and reliable models. This review will discuss current and emerging preclinical models (Table 1), aiming to facilitate a better understanding of the advantages and limitations of each methodology in order to better inform effective clinical trial design for immunotherapy in gynecologic cancers.

## 3. Preclinical Models of Gynecologic Malignancies

### 3.1. Patient-Derived Xenograft (PDX) Models

Patient-derived xenograft (PDX) models utilize implantation of patient tumor tissue or ascites tumor cells into mice and attempt to replicate a human tumor. PDX models are commonly used to study tumor biology, resistance to chemotherapy, clinical correlations and applications, as well as challenges and limitations [61]. For ovarian and fallopian tube cancers, xenografts develop similar metastatic patterns to humans when implanted in the peritoneal cavity and have been utilized to assess responses to chemotherapy regimens [62]. Liu et al. generated PDX models from tumor cells isolated from the ascites or pleural fluid of patients, providing a platform for proof-of-concept for efficacy and biomarker studies [63]. Depreeuw et al. characterized 24 endometrial cancer PDX models by engrafting fresh tumor tissues from patients and evaluated PI3K and MEK inhibitor treatments based on tumor mutations [64]. Other studies have used PDX models to monitor invasion and metastasis patterns [65], including orthotopic luciferase-tagged xenografts for endometrial cancer and models for cervical dysplasia and cancer [66,67,68]. From a translational standpoint, PDX models hold promise for high-throughput drug testing for patient-specific tumors, based on the rapid uptake of small quantities of human tumor tissue and high rates of engraftment; however, this application is still in its infancy. Although the advantage of studying human cells and tissues in vivo is significant, there are notable limitations. First, there may be genomic instability of these tumors; this complicates the ability to accurately model treatment responses in humans and may represent a potential confounder [69,70]. Human stroma is also replaced by murine stroma over time, limiting the application in investigating the TME [71]. This is of particular importance when targeting angiogenesis. Anti-VEGF agent bevacizumab, used clinically in gynecologic oncology, is often also used in preclinical models. However, this is a human monoclonal antibody: when murine stroma replaces human xenograft tissue, such agents may not act effectively against murine angiogenic factors [72]. This may allow for tumor support and growth that does not authentically represent human growth patterns: cross-species targeted therapy has been suggested as a possible intervention [73]. As the impact of anti-VEGF agents on TME interactions has expanded, the importance of accurately recognizing potential confounders such as host stroma angiogenesis must not be overlooked. Lastly, using immunodeficient host mice inherently limits the translatability of research on the complex TME and immune system interactions under consideration [62,71].

### 3.2. Syngeneic Murine Models

Syngeneic mouse models for gynecologic cancer utilize in vitro tumorigenesis with subsequent tissue reimplantation into mice [74,75]. The ID8 parental murine OC line provided valuable insight into tumor pathophysiology, including angiogenesis and the importance of VEGF in tumor growth [76]. Preclinical studies using this model have been used to investigate novel treatments, including the synergistic effects of vaccinia virus (VV) and anti-PD-1 therapy and C-X-C chemokine receptor type 4 (CXCR4) antagonist AMD3100 and anti-PD-1 combination therapy [54,55]. The syngeneic ovarian OV2944-HM-1 cell line (also known as HM-1) has also been used to assess immune cell recruitment and interactions, specifically MDSCs (myeloid-derived suppressor cells) and its influence on immune escape [77,78,79]. Notably, HM-1 may be subcutaneously injected into mice, compared to largely intraperitoneal ID8 models. As illustrated by experiments by Horikawa et al. regarding anti-VEGF therapy resistance, this heterogeneity in models provides a useful opportunity to compare tumor growth and microenvironment patterns within ovarian cancer models [80]. In endometrial carcinoma, a syngeneic model found that the PoleP286R allele in the endometrium enhances anti-tumor immune response and tumor regression, suggesting that POLE-driven cancers may be more sensitive to ICB [81]. However, these findings have not been translatable in clinical trials.

Genetic editing in syngeneic murine models, via the CRISPR (clustered regularly interspaced short palindromic repeats) and Cas9 (CRISPR-associated nuclease 9) (CRISPR–cas9) system, further develops the syngeneic murine model by approximating genetic mutations noted in human tumors. The system creates specific “knockdown” models that eliminate tumor suppressor genes or promote oncogenes, better replicating the genetic expression of the tumor of interest [82]. For example, OC frequently harbors a *p53* mutation. Via CRISPR/cas9 editing, *p53* knockout murine models can be created and are often used in preclinical research. Ghaffari et al. investigated the effects of carboplatin, STING agonist, and anti-PD-1 therapy combinations in the ID8 model harboring *p53* knockout (ID8p53^−/−^) [56]. Treatment with STING agonist modulated the TME towards a T_H_1-type immune response, and combination carboplatin + STING agonist + anti-PD-1 resulted in the best survival [56]. This technology can also be used to study the impact of various tumor mutations on the TME, as assessed in a panel of knockdown models developed by Iyer et al.; this team has also worked on developing syngeneic murine organoids using similar technology [83,84]. In cervical cancer, CRISPR/Cas9 targeting of HPV (human papilloma virus) and anti-PD-1 had synergistic anti-tumor effects [59]. Such studies provide examples of the disease mechanism processes that can be elucidated via syngeneic mouse models. However, they are inherently limited by being purely murine models without human cells. This prevents their translatability to human responses and TME.

### 3.3. Transgenic Murine Models

Transgenic murine models, namely the *MISIIR-Tag* mouse lineage, were developed to more accurately replicate the progression of OC. These mice form poorly differentiated, bilateral ovarian tumors that metastasize to the omentum and form ascites, similarly to human ovarian cancers [85]. This lineage is used in preclinical research of both chemotherapy and novel immunotherapy agents. However, the female mice from this model are infertile, posing a challenge in establishing a stable line [85]. In cervical cancer, K14E6/K14E7 double transgenic mice have been used to study recurrence and resistance to progestin therapy [86]. In endometrial cancer, transgenic mice with PTEN (Phosphatase and tensin homolog), p53, and mitogen-inducible gene 6 (MIG-6) knockouts have been developed to investigate common mechanisms in the development of that malignancy [87]. Fluorescence technology has been incorporated to allow in vivo monitoring of tumor progression in preclinical models [88], although recent research shows that fluorescent proteins may interfere with the immune response [89,90]. Overall, transgenic mice are useful in studying gynecologic malignancy development, especially with respect to recognized genetic mutations, but they are also limited by the genetic homogeneity inherent to the model’s development.

### 3.4. Humanized Murine Models

Humanized mice represent an in vivo model with significant potential in gynecologic oncology immunotherapy research. Immunodeficient mice are engrafted with human tumors and injected with human hematopoietic stem cells. Similar to PDX models, tumor growth follows a process faithful to that seen in humans. However, humanized mice also attempt to incorporate the immune response, including B and T cells [91], and therefore have the potential to recreate the TME while allowing systemic observation and manipulation, with improved fidelity to humans [92]. Humanized mice have been utilized in OC: a model developed with CAR-T cells demonstrated complete tumor regression [57], and monoclonal antibody immunotherapy in a separate model showed that anti-CCR4 enhanced the anti-tumor immune response by hindering the infiltration of Tregs into the TME [58]. Unfortunately, endometrial cancer humanized mouse models have not yet successfully been established [87], representing an area of unmet need.

The humanized mouse model has limitations that may prevent faithful representation of the TME. These include Graft Versus Host Disease due to persistent mouse immune response to human immune cells, difficulties in transgenic HLA (human leukocyte antigen) expression, and durable production of full human hematopoietic cell lines [93]. Additionally, this model requires significant financial investment: each mouse can cost hundreds of dollars.

## 4. Shifting Away from Murine Models

### 4.1. Co-Culture and Two-Dimensional (2-D) Models

Murine model limitations have opened the door to the optimization of more clinically relevant platforms. Co-culture and 3-D models are increasingly popular ways to study patient-derived tumor characteristics, including how cancer cells interact and communicate with the surrounding environment to regulate disease progression. These models aim to bridge the gap between oversimplified cell cultures and animal models.

Two-dimensional co-culture models plate one cell type into an adherent cell culture dish with the consecutive plating of a different cell population. These models have been used to study the T-cell killing of cancer cells. For example, a TICS (tumor-immune co-culture system) studying the human OC intrinsic mechanisms regulating lymphocyte activation in response to ICB identified a cross-resistance model, suggesting that acquired chemotherapy resistance may confer ICB resistance and that agents restoring HLA expression represent a partner for immunotherapy in chemotherapy-relapsed OC patients [94]. Another example of a co-culture model is the transwell assay. This method can incorporate multiple cell types and is often used to assess the secretion of soluble factors, cell migration, and invasion through an extracellular matrix. For example, in OC-involved omental tissue, transwell assays have been used to identify potential targets to limit tumor cell invasion and migration [95,96]. The transwell system showed that gamma-delta T lymphocytes on human OC stem-like cells (SK-OV-3) can efficiently kill cancer stem cells through IL-17 production, highlighting a promising immunotherapy [97]. In endometrial cancer, the transwell model showed that adiponectin from mesenchymal stem cells enhanced cancer proliferation [98]. Co-culture models have demonstrated that, when grown together, human cervical epithelial and fibroblast cells increase local chemokine expression [99]; the model itself, in this case, provided insight into the complexity of the TME and local inflammatory pathways. Immunotherapy works well in a “hot” TME, and the increase in the expression of certain chemokines suggests sensitivity to immunotherapy. Co-culture models, therefore, show potential via their ability to reconstruct components of the human TME and isolate interactions involved in ICB.

### 4.2. Three-Dimensional (3-D) Models

Some co-culture models utilize tumor and stromal cells in a matrix to create a 3-D system, improving traditional 2-D methods by incorporating tissue architecture into these models and allowing investigation of the role of drug penetrance and distribution [100]. These models allow researchers to observe and manipulate a medium approximating the in vivo condition. Benefits include having human stromal and immune components in conjunction with tumor cells to approximate the TME, allowing research into cellular interactions. This is potentially advantageous for immunotherapy, where complex molecular signaling and its impact on the TME are of particular interest. The 3-D multi-cellular spheroid model can represent in vivo micrometastasis via the creation of a matrix incorporating human stromal cells or replicate ascites via the “hanging drop” method [95,101,102,103], particularly pertinent in OC. The spheroid structure allows approximation of in vivo physiology, incorporating consideration of 3-D growth components such as drug penetration, metabolism, chemokine interactions, and O_2_ diffusion [104]. Human immune cells can be included, increasing their potential in immunotherapy research, as demonstrated by Wan et al. and their work on PD-1/PD-L1 ICB, incorporating NK cells into patient-derived organoids [60]. Shigeta et al. have used the organoid model to investigate PI3K/AKT pathways in ovarian clear cell carcinoma, demonstrating its utility in high-throughput treatment for rare cancers [105]. In endometrial cancers, 3-D spheroids were used to analyze drug efficacy, cell viability, and proteomics [106]. Studies have also used patient-derived endometrial cancer organoids as a high-fidelity 3-D model preserving the TME and tumor–stromal interactions [107]. These models utilize limited quantities of human tissue to replicate the original physiology.

### 4.3. Models in Development

Moving forward, one of the most promising approaches to reproducing tumor structure and microenvironment is tumor tissue engineering (TTE) bioreactor systems. Bioreactors are closed systems that allow controlled experimental conditions, typically including a tank, pumps, and sensors, with human tumor tissue, scaffolds, and other components of the TME [108]. Scaffolds are a key component, functioning as a bioreactive extra-cellular matrixproviding mechanical support for cells in order to promote adhesion and/or motility, stimulate required differentiation, and maintain cell functionality while incorporating perfusion and gas exchange components to assess cellular growth and treatment response [108]. A perfusion-based bioreactor system using TNBC (triple-negative breast cancer) patient tissue found that ICB increased immune activation and proliferation followed by cancer cell death, highlighting the use of bioreactors for the preclinical evaluation of conventional and immune-mediated treatments [109]. OC bioreactors are in development but remain investigational [110].

When compared to murine models that may require immunosuppression or incorporate murine tissue responses, co-culture systems may also be helpful for immunotherapy by limiting potential confounders. These models also use human tissue directly. Financially, co-culture systems may be a feasible preliminary research model for novel therapeutics targeting the TME prior to attempting more complex and resource-heavy in vivo studies. However, these systems are inherently limited by being in vitro models. Additionally, these models are synthetically constructed, and incorporation of all components of the TME must therefore be consciously performed. As the roles of fibroblasts, vascular tissue, and other cells are recognized, more simplistic systems may not replicate in vivo complexity; in addition, potentially valuable TME factors that are not yet recognized may be overlooked. The scale of co-culture models—either as droplets or in vitro matrices—also limits the observation and manipulation of larger tissue growth. The perfusion-based bioreactor, in particular, lacks vascular factors [111] that may play an important role in tumor growth and immunotherapy, and each unit has a short time-span of viability. Overall, the utility and translational value have yet to be fully defined for these models, but their potential represents an actively developing area for preclinical tissue modeling.

## 5. Conclusions

Despite the significant need for novel therapies in gynecologic oncology treatment, breakthroughs have been modest. Immunotherapy is an emerging field with multiple pathways under active investigation, but rather than attempting numerous permutations of new agents and regimens, a thorough understanding of expected outcomes and limitations in preclinical models will improve the efficiency of investments in preclinical research and contribute to meaningful clinical trial outcomes. However, this requires careful study development and a thorough understanding of each model and its ideal role in preclinical immunotherapy research, in order to effectively analyze and interpret findings. As seen in this review, each model (Table 2) has characteristics that influence its translational applicability: effective study design will require multi-disciplinary pharmaceutical, researcher, and clinician collaboration. Utilizing appropriate preclinical models will be integral to identifying the interventions with the highest translational potential and developing impactful, efficient science that will improve patient outcomes in gynecologic malignancies.

## Figures and Tables

**Table 1 cancers-13-01694-t001:** Promising preclinical models utilizing immunotherapy interventions in gynecologic malignancies.

Model		Description	Immunotherapy	References
Syngeneic	ID8	Murine ovarian cancer	Vaccinia virus (VV) + anti-PD-1 CXCR4 agonist (AMD3100) + anti-PD-1	[54,55]
	ID8p53^−/−^	CRISPR/Cas9 *TP53* murine ovarian cancer	Carboplatin + STING agonist + anti-PD-1	[56]
Humanized	SKOV-3 IGROV-1, OVCAR-5, and OVCAR-8	Human ovarian cancer	CAR-T cells Anti-CCR4	[57,58]
	SiHa	Human cervical cancer	CRISPR/Cas9 HPV + anti-PD-1	[59]
Organoids	Patient derived	Human ovarian cancer	Anti-PD-1	[60]

**Table 2 cancers-13-01694-t002:** Summary of preclinical models used in gynecologic oncology research.

Model		Description	References
Syngeneic	ID8 ID8p53^−/−^ HM-1	Murine ovarian cancer	[54,55,56,76,78,79,83]
	Conditional *Pole^P286R/+^* knock-in mice	Murine endometrial	[81]
Transgenic	*MISIIR-Tag*	Spontaneous murine ovarian cancer	[85]
	K14E6/K14E7—double transgenic	Murine cervical cancer	[86]
		Endometrial	[87]
Humanized		Ovarian	[57,58,91,92,93]
	SiHa	Cervical	[59]
Organoid/3-D		Ovarian Organoids	[84,95,101]
		Ovarian Spheroids/3-D	[60,102,103,104]
		Endometrial	[106,107]

## Data Availability

The data presented in this study are available on request from the corresponding author.

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
