# Peer review of "Assessing Preclinical Research Models for Immunotherapy for Gynecologic Malignancies"

_cancers, 2021, doi:10.3390/cancers13071694_

Round 1

Reviewer 1 Report

The review article on "pre-clinical research models for immunotherapy for 2
gynecologic malignancies" comprehensive and covers various aspects of benefits of research model available for gynecologic malignancies.  I recommend this article for publication with minor concerns below.

i) The table listed in the review articles is of poor resolution and difficult to read the contents in the table.

Author Response

i) The table listed in the review articles is of poor resolution and difficult to read the contents in the table.

Response: We appreciate this feedback. The tables appear readable at the time of submission, however if processing creates an issue, we have enhanced resolution images of both tables.

Reviewer 2 Report

The authors well-demonstrated a comprehensive review about pre-clinical  in vitro and in vivo experimental models of immunotherapies for ovarian cancers. The review seems to be appropriate to publish in Cancers, after some points are improved.

-In the sentence of anti-VEGF related therapy, there is a controversial  problem of inhibition of VEGF in preclinical setting. Which is good approach of anti-VEGF examination with anti-human VEGF antibody or with anti-mouse (host) one in xenograft mouse model ? Mouse vascular endothelium can be stimulated human VEGF ?  Please refer this problem and previously published papers in Reference or Discussion.

-In table 2 and in 3.2 Syngeneic Murine Models, OV2944-HM-1 mouse ovarian cancer cell line is also used for several preclinical models of ovarian cancer, however the authors did not refer to HM-1.
(https://cellbank.brc.riken.jp/cell_bank/CellInfo/?cellNo=RCB1483).

Author Response

In the sentence of anti-VEGF related therapy, there is a controversial  problem of inhibition of VEGF in preclinical setting. Which is good approach of anti-VEGF examination with anti-human VEGF antibody or with anti-mouse (host) one in xenograft mouse model ? Mouse vascular endothelium can be stimulated human VEGF ?  Please refer this problem and previously published papers in Reference or Discussion.

  • This is an excellent point; we appreciate the reviewer’s comment and have added this discussion with references to section 3.1

In table 2 and in 3.2 Syngeneic Murine Models, OV2944-HM-1 mouse ovarian cancer cell line is also used for several preclinical models of ovarian cancer, however the authors did not refer to HM-1.
(https://cellbank.brc.riken.jp/cell_bank/CellInfo/?cellNo=RCB1483).

  • Thank you for this feedback; discussion of HM-1 has been added to both the table and pertinent section, with published papers referenced.

Reviewer 3 Report

The review is well described and can be accepted with following minor modifications.

Comments:

Table 1/2 is not clear/visible in the PDF.

Section of Targeted molecular therapy/Immunotherapy should be expanded with recent research.

Addition of a couple of Schematic representations will enhance the impact of the Review.

What about other immunotherapies (besides PD-L1) in Gynecologic malignancies? Is there any ongoing Clinical trials?

  The manuscript, I reviewed is novel in the specified area. The manuscript is well written with clear text and easy to read. However, tables are not visible as indicated in the comments. It's a Review and conclusions are fine. The clinical relevance can be described. 

Author Response

Section of Targeted molecular therapy/Immunotherapy should be expanded with recent research.

  • This section has been expanded, notably with recent developments in PI3K/AKT/mTOR research in gynecologic malignancies.

Addition of a couple of Schematic representations will enhance the impact of the Review.

  • We aimed to keep figures succinct with the overview Figure 1, which graphically demonstrates the models discussed in a comparative format.

What about other immunotherapies (besides PD-L1) in Gynecologic malignancies? Is there any ongoing Clinical trials?

  • CTLA4 axis interactions are also under investigation in gynecologic malignancies. This is discussed in section 1 (page 4, above Table 1) of the manuscript. Combination PD-L1 and antiCTLA4 therapies are also discussed in this section, including the recent/ongoing lenvatinib/pembrolizumab trials series in endometrial cancer. Anti-CTLA4/PARP combination therapy has been added, and extensive discussion regarding the PI3K/AKT/mTOR pathway has been added to sections 2.1 and 2.2. 

Reviewer 4 Report

Well-written, easy to read, overview of different models applied in context of gynaecological cancers. Some minor issues remain

  • Title does not cover a third of the manuscript, in particular chapter 2. I suggest to broaden the title.
  • Can the authors explain why clinical successes with immunotherapy for gynaecological cancers are lagging behind compared to other malignancies?
  • Hormonal-based therapies are hardly described in the manuscript
  • Resolution figures and tables should be improved. Currently not very readable.
  • Check space usage across the manuscript, sometimes space is missing, sometimes double spaces.
  • Several recent publications on ovarian organoids are missing in the text description
  • Table 2: I would split organoid vs spheroid model references.

Author Response

Title does not cover a third of the manuscript, in particular chapter 2. I suggest to broaden the title.

  • We believe that chapter 2 is integral to understanding the complexity of immunotherapy preclinical models research and therefore relates to the title. With the additions recommended by reviewers, we hope that this now appears more cohesive.

Can the authors explain why clinical successes with immunotherapy for gynaecological cancers are lagging behind compared to other malignancies?

  • This section has been added to the introduction, with reference.

Hormonal-based therapies are hardly described in the manuscript

  • This was exempted in order to maintain the focus on preclinical and translational research in molecularly targeted pathways and immunotherapy; although hormonal therapies are of great interest in gynecologic oncology, comprehensive overview of this research including model systems would merit its own discussion. However, the reviewer’s point is well taken, and pertinent discussion of clinical trials and pathways has been added (discussion of aromatase inhibitors and everolimus specifically.)

Resolution figures and tables should be improved. Currently not very readable.

  • These have been edited.

Check space usage across the manuscript, sometimes space is missing, sometimes double spaces.

  • This has been remedied.

Several recent publications on ovarian organoids are missing in the text description

  • Our discussion on organoids includes multiple references from 2021, from the Iyer, Wan, Zhang, and Shigeta teams.

Table 2: I would split organoid vs spheroid model references.

  • This has been performed, and references updated to include HM-1 syngeneic line as well.